

# Clustering-based Failed goal Aware Hindsight Experience Replay

Taeyoung Kim[1], Taemin Kang[2], Haechan Jeong[1] and Dongsoo Har[1]

[1] CCS Graduate School of Mobility, Korea Advanced Institute of Science & Technology, Daejeon, Republic of South Korea

[2] The Robotics Program, Korea Advanced Institute of Science & Technology, Daejeon, Republic of South Korea

## ABSTRACT

In a multi-goal reinforcement learning environment, an agent learns a policy to perform tasks with multiple goals from experiences gained through exploration. In environments with sparse binary rewards, the replay buffer contains few successful experiences, posing a challenge for sampling efficiency. To address this, Hindsight Experience Replay (HER) generates successful experiences, named hindsight experiences, from unsuccessful ones. However, uniform sampling of experiences for the process of HER can lead to inefficient scenarios of generating hindsight experience. In this paper, a novel method called Failed goal Aware HER (FAHER) is proposed to enhance sampling efficiency. This method considers the properties of achieved goals with respect to failed goals during sampling. To account for these properties, a cluster model is used to cluster episodes in the replay buffer, and experiences are subsequently sampled in the manner of HER. The proposed method is validated through experiments on three robotic control tasks from the OpenAI Gym. The experimental results demonstrate that the proposed method is more sample-efficient and achieves improved performance over baseline approaches.

## INTRODUCTION

Reinforcement learning (RL) is a powerful learning approach, a branch of machine learning, where an agent learns sequential actions to complete a specific task, particularly in environments with uncertain or delayed rewards. Within the RL approach, the agent learns to take actions that maximize cumulative rewards. The actions executed by the agent are outputs of a policy function, with the state serving as the input. The use of deep neural networks for approximating policy functions has catalyzed remarkable progress in RL, enabling its application across various domains, such as video games (*Mnih et al., 2015*; *Vinyals et al., 2019*; *Perolat et al., 2022*), smart manufacturing (*Dittrich & Fohlmeister, 2020*; *Liu et al., 2023*), autonomous vehicle (*Folkers, Rick & Büskens, 2019*; *Jung & Oh, 2022*; *Carrasco & Sequeira, 2023*), and robotics (*Seo et al., 2019*; *López-Lozada et al., 2021*; *Kim et al., 2021*; *Gu et al., 2023*).

Corresponding author
Dongsoo Har, dshar@kaist.ac.kr

Real-world tasks often involve multiple goals. For example, when walking, an agent navigates to various target destinations. Similarly, in an object placement task, the agent positions an object at different locations on a table surface. To address such tasks with multiple goals, the RL framework is extended to multi-goal RL (MGRL) (*Plappert et al., 2018*). The key difference from general RL is that the policy is conditioned on the goal. This policy is termed a goal-conditioned policy, leading to the alternate terminology of goal-conditioned RL for MGRL.

In both general RL and MGRL, the training dataset is experiences obtained from the exploration, which means the trial-and-error interactions of the agent within the environment. Each experience is comprised of the state, goal (in the MGRL case), action, reward, and next state. Since the loss function for training is based on the reward, properly setting the reward is a crucial problem. The reward must be carefully designed and shaped in accordance with the characteristics of the environment, which requires domain expertise and carries the risk of inaccuracies. Reward design is such a critical issue that it constitutes its own research area (*Hadfield-Menell et al., 2017*; *Silver et al., 2021*). To circumvent these risks and the need for domain expertise, the ideal approach is to train the agent using binary rewards. In the binary reward setting, the agent receives a reward of 0 for succeeding at the task and -1 for failing.

Since the agent takes random actions throughout most of the exploration, valuable experiences are scarcely obtained in environments with binary rewards. It is inefficient for these rare valuable experiences to be used only once as training data. This can lead to prolonged training times or even failure to learn the task. To address this issue, the experience replay (ER) technique is proposed by *Lin (1992)*. The ER technique stores the experiences obtained through exploration in a database called a replay buffer. During training, a mini-batch consisting of the experiences sampled from the replay buffer is utilized to train the policy. A crucial consideration is that experiences generated consecutively can exhibit strong correlations when the agent learns from them in sequence. For instance, if the agent takes similar actions across successive states, this can induce bias in the learning process. The ER technique serves to alleviate such bias by breaking the temporal coherence among the experiences used for training.

In environments with huge state spaces, such as 3D robotic control tasks, the use of binary rewards often leads to sparse reward signals. This sparseness of rewards is particularly pronounced in the MGRL framework, exacerbating the learning difficulty. Due to the sparsity of rewards, the number of successful experiences is typically low, resulting in a low proportion of successful experiences within the replay buffer. Consequently, mini-batches containing a few successful experiences are created and used for training. To address this issue, the Hindsight Experience Replay (HER) technique (*Andrychowicz et al., 2017*) generates successful experiences, termed hindsight experiences (HEs), from the experiences in the replay buffer, thereby increasing the sampling efficiency. HER operates at the episode level, where an episode refers to a sequence of experiences until the agent achieves a goal or fails. An HE is generated by substituting the original goal of an episode with an achieved goal (AG) during that episode. The AG differs from the original goal and refers to an outcome or state reached by the agent during an episode, regardless of whether the episode

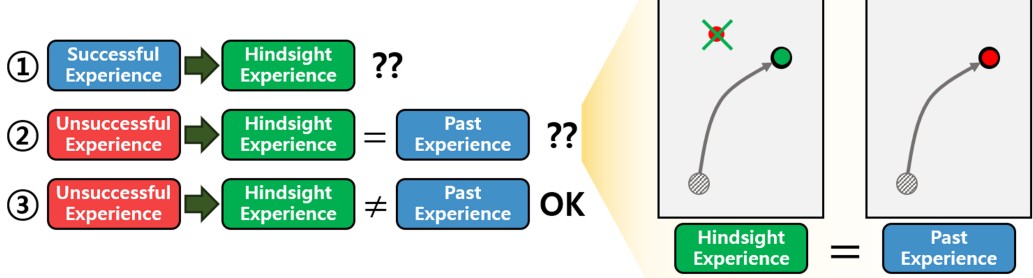

**Figure 1** **Illustration depicting the motivation behind the proposed method.** On the left side, three scenarios of generating HEs are depicted. The green arrows indicate the transformation of experiences in the replay buffer into the HEs. The question marks suggest that inefficiency is present in each scenario. On the right side, an example of the second scenario is illustrated. The task involves moving an object to the goal. In the illustration, the black dot represents the object and the grey line depicts the trajectory of the object. The red dot and the green dot denote the original goal and the AG at the last timestep of the episode, respectively.

is successful or unsuccessful. When combined with off-policy RL algorithms such as Deep Q-Network (*Mnih et al., 2015*), Deep Deterministic Policy Gradient (DDPG) (*Lillicrap et al., 2015*), and Soft Actor-Critic (*Haarnoja et al., 2018*), HER enables learning complex tasks with sparse binary rewards. HER can be integrated with a variety of methodologies, as demonstrated in *Vecchietti, Seo & Har (2020)*; *Zhao & Tresp (2018)*; *Yang et al. (2023)*; *Huang & Ren (2023)*; *Sanchez et al. (2024)*, to enhance performance.

HER generates HEs by uniform sampling from the replay buffer containing both successful and unsuccessful experiences. However, generating HEs without considering the characteristics of the experiences in the replay buffer is inefficient. As illustrated in Fig. 1, the process of generating HEs can be categorized into three scenarios: (1) generating HEs from successful experiences, (2) generating HEs from unsuccessful ones that match past experiences, and (3) generating HEs from unsuccessful ones that do not match past ones. The past experiences mean experiences that the current RL policy has gained. In the first scenario, the generated HEs are identical to the past successful experiences, resulting in an inefficient use of computing resources. The second scenario involves HEs that are also identical to the past ones. In this case, similar experiences are repeatedly used in the learning process, reducing data diversity and increasing the risk of overfitting. The third scenario represents the complement of the previous two. It generates new HEs that succeed in achieving goals where the current RL policy has failed. Consequently, the third scenario is more efficient compared to the others.

This paper demonstrates that considering the properties of the AG regarding failed goals (FGs) when sampling experiences can improve the efficiency of HER. The FG is defined as the original goal of an unsuccessful episode. From this perspective, the Failed goal Aware Hindsight Experience Replay (FAHER) is proposed. In FAHER, a cluster model is employed to consider the properties of the AGs regarding the FGs. The uniform sampling of the original HER is combined with the cluster model. By the cluster model, the replay buffer is divided into partitions called clustered buffers, followed by uniform sampling

from each clustered buffer to generate HEs. The cluster model, whose parameters are fit to the FGs, assigns a cluster index to the AG of each episode. Episodes in the replay buffer are allocated into cluster buffers based on their assigned indices. The core idea of driving this episode clustering is to empower the agent to extract richer insights from unsuccessful episodes, particularly where the AGs closely align with the FGs. The main contributions of this paper are as follows.

1. A clustering-based extension of HER is proposed. The key feature is the integration of episode clustering into the HER process. The episode clustering is based on the properties of the AGs regarding the FGs. During the creation of mini-batches, episodes with AGs that closely resemble goals currently challenging for the RL policy to achieve are sampled more frequently.

2. A method to utilize a cluster model on the experiences stored within the replay buffer to enhance HER is presented. This method involves fitting the cluster model parameters to the FGs and clustering the AGs. The concept of using FGs is to reduce the sampling of successful experiences during the generation of HEs.

3. The proposed method is evaluated on the Fetch environments of OpenAI Gym (*Brockman et al., 2016*) to demonstrate the performance improvement of HER by utilizing the cluster model with AGs and FGs. To conduct additional experiments, three variations of the Slide task are constructed.

4. An analysis is presented for comparative experiments and ablation studies. The comparative experiments encompass experiments to show the compatibility with other sampling algorithms. The ablation studies aimed at demonstrating characteristics of the key methodological components of the proposed method.

The remainder of this paper is structured as follows. 'System Modeling' describes the concepts of RL and MGRL, HER, variants of HER, K-means, and terminology clarification. In 'Proposed Method', the proposed method is introduced in detail. In 'Experiments', experimental environments and the results of comparison experiments and ablation studies are presented. 'Conclusion' concludes this paper.

## SYSTEM MODELING

In this section, the concepts of RL and MGRL, HER, variants of HER, K-means clustering algorithm, and terminology clarification are presented.

### Reinforcement learning

Reinforcement learning (RL) is a framework where an agent learns to make decisions by interacting with an environment to maximize cumulative rewards. At each timestep $t$, the agent observes a state $s_t \in S$ and selects an action $a_t \in A$ according to a policy $\pi : S \to A$. The environment responds to this action by transitioning to a new state $s_{t+1}$ based on a state transition probability $p(s_{t+1}|s_t, a_t)$ and provides a reward $r_t = r(s_t, a_t)$. The agent's goal is to learn a policy $\pi$ that maximizes the expected sum of future rewards, often referred to as the return.

The agent's learning process is guided by experiences $e_t = (s_t, a_t, r_t, s_{t+1})$, which are stored in a replay buffer for training. This approach allows the agent to use past experiences for training, improving the sample efficiency of learning.

## Multi-goal reinforcement learning

Multi-goal reinforcement learning (MGRL) extends the RL framework by introducing goals that the agent aims to achieve within a given task. The policy takes both the state and the goal information as input, being referred to as a goal-conditioned policy (*Schaul et al., 2015*). The reward function is a function of the state, goal, and action.

At the beginning of each episode, the environment provides an initial state $s_0 \in S$ and a goal $g \in G$, with the goal $g$ remaining fixed throughout the episode. The state comprises the observation $o$ and the AG $ag$. In environments with an object, the AG represents the state of the object. At each timestep $t$, the agent takes an action $a_t \in A$ according to the policy $\pi : S \times G \rightarrow A$, given the current state $s_t$ and the goal $g$ as input. The environment is affected by the action $a_t$ and returns a reward $r_t = r(s_t, g, a_t)$ and the next state $s_{t+1}$. The next state is determined by the state transition probability $p(s_{t+1}|s_t, a_t)$. The interaction of the agent with the environment continues until a terminal state is reached. The experiences $e_t$ obtained through exploration are represented as a 5-tuple $(s_t, g, a_t, r_t, s_{t+1})$.

## Hindsight experience replay

Hindsight Experience Replay (HER) (*Andrychowicz et al., 2017*) addresses the issue of sparse rewards commonly encountered in MGRL. When positive reward experiences are rare, HER improves sample efficiency by reinterpreting unsuccessful episodes as successful ones, thereby enabling the agent to learn from its failures.

HER generates an HE by replacing the original goal $g$ with a hindsight goal $g^h$, which is an AG from the same episode. By recalculating the reward with respect to $g^h$, *i.e.*, $r_t^h = r(s_t, g^h, a_t)$, the original experience $e_t = (s_t, g, a_t, r_t, s_{t+1})$ is transformed into a new hindsight experience $e_t^h = (s_t, g^h, a_t, r_t^h, s_{t+1})$. This process allows the agent to learn from outcomes it was able to achieve, rather than solely focusing on the initially specified goal, thus improving learning efficiency in environments with sparse rewards.

## Variants of hindsight experience replay

Several variations of HER have been proposed to address its limitations, particularly in improving sample efficiency within environments with sparse rewards. These variants introduce different mechanisms to enhance HER's ability to handle complex tasks more effectively. This subsection summarizes the key approaches of 4 variants.

*Zhao & Tresp (2018)* introduces energy-based prioritization (EBP) to HER, which incorporates a trajectory energy function to evaluate the difficulty of tasks based on the physical energy of the agentâôs interactions. This variant prioritizes trajectories with higher energy levels, focusing learning on more challenging but achievable tasks. Multi-step HER (*Yang et al., 2023*) addresses the inherent bias in standard HER by relabeling multiple consecutive transitions within a trajectory rather than individual transitions. This multi-step relabeling increases the number of non-negative learning signals, thereby improving the learning process in sparse reward settings. Robust Model-based HER (*Huang & Ren,*

*2023*) extends HER with a predictive dynamics model that forecasts future states. This approach introduces foresight relabeling, where the predicted future states are treated as achieved goals. *Sanchez et al. (2024)* proposes Q-switch Mixture of Primitives HER, which improves sample efficiency by leveraging previously learned primitive behaviors from simpler tasks. These primitive policies are reused when tackling more complex tasks, reducing exploration time and allowing the agent to transfer knowledge across tasks.

In contrast to these existing approaches, the proposed method addresses the problem of inefficient sampling in HER by focusing on the properties of FGs and AGs. This paper introduces a clustering-based framework that groups experiences. This approach differs from the above variants by explicitly using the characteristics of unsuccessful episodes. Through experiments integrating the proposed clustering approach with HER-EBP, this paper shows that the proposed method can work alongside existing HER variants to further enhance sample efficiency.

## K-means clustering

The K-means clustering (*McQueen, 1967*) is a widely adopted unsupervised clustering algorithm. It partitions the input data into $k$ distinct clusters. The number of clusters, $k$, is a user-specified parameter. The algorithm is initialized by randomly selecting $k$ centroids, the centers of the clusters. Each data point is assigned to its nearest centroid. The centroids are subsequently updated to minimize the average squared distance between the data points and their assigned centroids. This process of assigning data points and updating centroids is performed iteratively until the centroids converge, at which point there are no further transitions. K-means clustering is suitable for general-purpose applications and scenarios involving a relatively small number of clusters. Therefore, in this paper, it is utilized to fit the cluster model parameters to FGs for clustering episodes in the replay buffer.

## Terminology clarification

This paper uses several key terms to describe different aspects of experiences and goals in HER. An illustrative example of HER is first provided, which will be referenced throughout the following explanations.

### Illustrative example

Consider a scenario in which a robot is tasked with placing an object at a target point *A* and ends up placing it at point *B*. If points *A* and *B* are identical/close, the experience is considered successful. However, if they differ significantly, the experience is considered unsuccessful. In such cases, HER reinterprets it by assuming that the point *B* was the intended target from the outset, thus transforming it into an HE.

### Key terms

- **Hindsight experiences (HEs)** refer to successful experiences created by HER from the experiences stored in the replay buffer.
- **Past experiences** refer to experiences gained by the current RL policy, representing experiences where the current policy can successfully achieve the goal. The proposed method aims to avoid generating hindsight experiences that are similar to past experiences, as this can reduce the efficiency of learning.

- **Original goal** represents the target that the agent was tasked with achieving, which in the example is point *A*.
- **Achieved goal** (**AG**) is the state or outcome the agent reaches based on its actions. In the example, the AG is point *B*, the location where the object was placed.
- **Failed goal** (**FG**) refers to the original goal in an unsuccessful experience, where the agent did not achieve its target. In the example, when the points differ, point *A* becomes the FG, representing a goal that the current RL policy was unable to achieve. The proposed method focuses on sampling experiences where the AG (point *B*) is similar to the FG.

## PROPOSED METHOD

This section details the proposed FAHER method. It employs a cluster model to enhance the sampling efficiency of HER. The cluster model clusters the episodes in the replay buffer when sampling experiences to create mini-batches.

For MGRL settings with sparse binary rewards, HER allows the agent to gain valuable positive feedback from unsuccessful experiences by generating HEs. An HE is generated by selecting one AG from the AGs of an episode as the hindsight goal and recomputing the rewards accordingly. However, as mentioned in the 'Introduction', the generation of HEs can be categorized into three scenarios, with the first two being less sample-efficient compared to the third one.

A method is proposed to increase the occurrence of the third scenario. The unsuccessful episodes of the third scenario are termed Potential-insights Episodes (PiEs). The PiEs are defined as episodes where the AGs, if set as the original goals, would be difficult to achieve under the current RL policy. In other words, PiEs hold the potential insights to generate HEs that represent successful experiences the current RL policy cannot produce itself. The AGs of PiEs are characterized by their proximity to past FGs. The proposed method focuses on the PiEs by making HER aware of the FGs and aims to improve HER.

The core idea of the proposed method is to introduce a clustering procedure into the original HER framework. The original HER framework consists of three uniform sampling processes: the first samples episodes from the replay buffer; the second samples an experience from each sampled episode; and the third samples experiences to be converted into HEs among the sampled experiences, as shown in the upper part of Fig. 2.

The first sampling process of HER can be redesigned such that PiEs are sampled more frequently, thereby enhancing the effectiveness of HER. To realize this concept, FAHER replaces the first sampling process with a sequential process of two procedures: clustering of episodes using a cluster model; and uniform sampling of episodes from the clustered episodes, as depicted in the lower part of Fig. 2.

The clustering procedure relies on a cluster model. As shown in Fig. 3, the cluster model parameters are fit to the FGs stored in the failed goal buffer (FGB). These FGs are collected during the training of the RL policy. Once the cluster model is fit, it assigns each episode in the replay buffer to a cluster based on its last AG, which is the AG at the end of the episode. The last AG is used because the success or failure is determined by comparing the

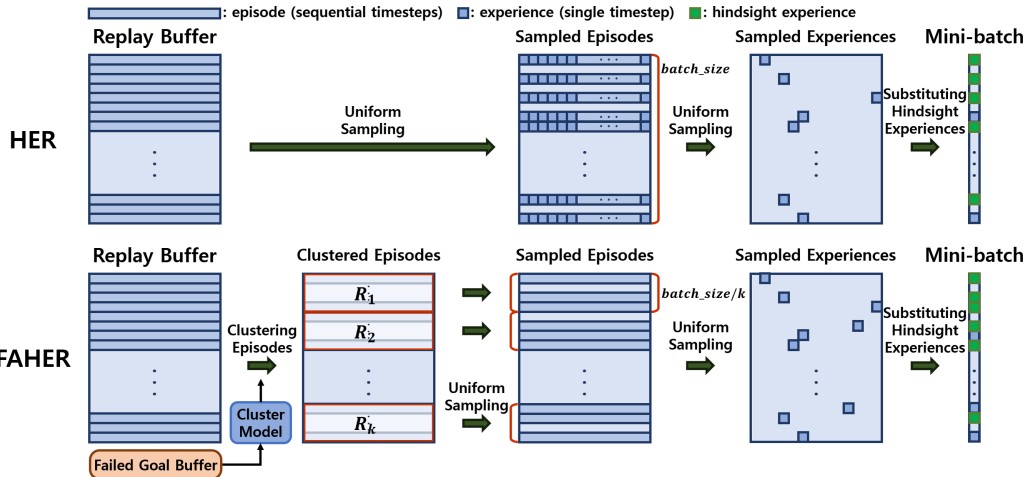

**Figure 2 Frameworks of HER and FAHER.** The upper row depicts the process of generating HEs. Episodes, which consist of experiences for sequential timesteps, are uniformly sampled three times to create a mini-batch. In contrast, the bottom row shows the enhanced process using a cluster model applied to FGs stored in the failed goal buffer. The episodes are first grouped into clustered buffers and sampled from each clustered buffer. This results in generating diverse and meaningful hindsight experiences. Annotations of $R_i$ and $k$ represent $i$th clustered buffer and the number of the clusters.

last AG with the original goal. Based on these cluster indices, the episodes are clustered into clustered buffers. The clustered buffers $R_i$ are subsets of the replay buffer, and their number thereof is equal to the number of clusters $k$.

From each clustered buffer, *batch_size*/$k$ episodes are uniformly sampled to form an episode batch containing *batch_size* episodes. This sequential process of clustering and sampling is referred to as a clustering-based sampling strategy. With the episode batch, the second and third uniform samplings are performed in the manner of HER.

The clustering model needs to be periodically updated to remain effective. This periodic update is a crucial feature of the proposed method, as using a cluster model fitted to outdated FGs can hinder the training of the RL policy. The cluster model is updated when the FGB is fully replenished with new FGs. Once the cluster model is updated, it reassigns cluster indices to all the episodes in the replay buffer. Episodes stored after the cluster model update are individually assigned cluster indices.

The key parameters for the periodic update of the cluster model are the number of FGs used for fitting and the update cycle. The first parameter is equivalent to the size of the FGB. These two parameters must be carefully determined. A small FGB size may not adequately represent the FGs of the current RL policy. A large FGB size increases computational costs and causes the use of outdated FGs from the past RL policy. In the case of a short update cycle, the frequently changing cluster model may not provide sufficient time for the RL policy to learn the current FGs. A long cycle can lead to wasted time after the RL policy has fully learned the current FGs. In the following section, ablation studies of these important parameters are presented.

---

**Algorithm 1** Clustering-based sampling strategy

---

**Given:** a replay buffer $R$, a cluster model $\mathbb{C}$

 1: Initialize a episode buffer $\mathcal{B}$
 2: Define $A$ as a set of the last achieved goals $ag_T$ of episodes in $R$
 3: Assign the cluster index to each $ag_T$ in $A$ by using $\mathbb{C}$
 4: **for** $i = 0, k\text{-}1$ **do**
 5:   Define a clustered buffer $R_i$ containing the episodes whose cluster index is $i$
 6:   Sample $\frac{batch\_size}{k}$ episodes from $R_i$
 7:   Store sampled episodes to $\mathcal{B}$
 8: **end for**

---

**Algorithm 2** Failed goal Aware Hindsight Experience Replay (FAHER)

---

**Given:** an off-policy RL algorithm $\mathbb{A}$, a cluster model $\mathbb{C}$, a clustering-based sampling strategy $\mathbb{S}_c$, a strategy $\mathbb{S}_g$ for sampling goals for replay, a reward function $r : S \times A \times G \to \mathbb{R}$

 1: Initialize neural networks $\mathbb{A}$, a cluster model $\mathbb{C}$, replay buffer $R$, and failed goal buffer $F$
 2: **for** epoch $= 1$, **K do**
 3:   **for** episode $= 1$, **M do**
 4:    Sample a goal $g$ and initial state $s_0$
 5:    **for** t $= 0$, **T** $- 1$ **do**
 6:     Select an action $a_t$ using the behavior policy from $\mathbb{A}$: $a_t \leftarrow \pi(s_t, g) + N_t$
 7:     Execute the action $a_t$ and observe a new state $s_{t+1}$ and $r_t = r(s_t, a_t, g)$
 8:     Store the experience $(s_t, g, a_t, r_t, s_{t+1})$ temporarily
 9:    **end for**
 10:    Store the experiences as an episode in $R$
 11:    **if** The episode is unsuccessful **then**
 12:     Store the last achieved goal in $F$
 13:    **end if**
 14:    Fit $\mathbb{C}$ to $F$
 15:    **for** i $= 1$, **N do**
 16:     Sample a set $\mathcal{B}$ of episodes from $R$ with $\mathbb{S}_c$
 17:     Sample a mini-batch $B$, a set of experiences, from $\mathcal{B}$
 18:     Sample a set of achieved goals $G$ with $\mathbb{S}_g$
 19:     **for** $g^h \in G$ **do**
 20:      $r_t^h = r(s_t, a_t, g^h)$
 21:      Substitute $g$ and $r_t$ in the experience by $g^h$ and $r_t^h$
 22:     **end for**
 23:     Perform one step of optimization using $\mathbb{A}$ and $B$
 24:    **end for**
 25:   **end for**
 26: **end for**

---

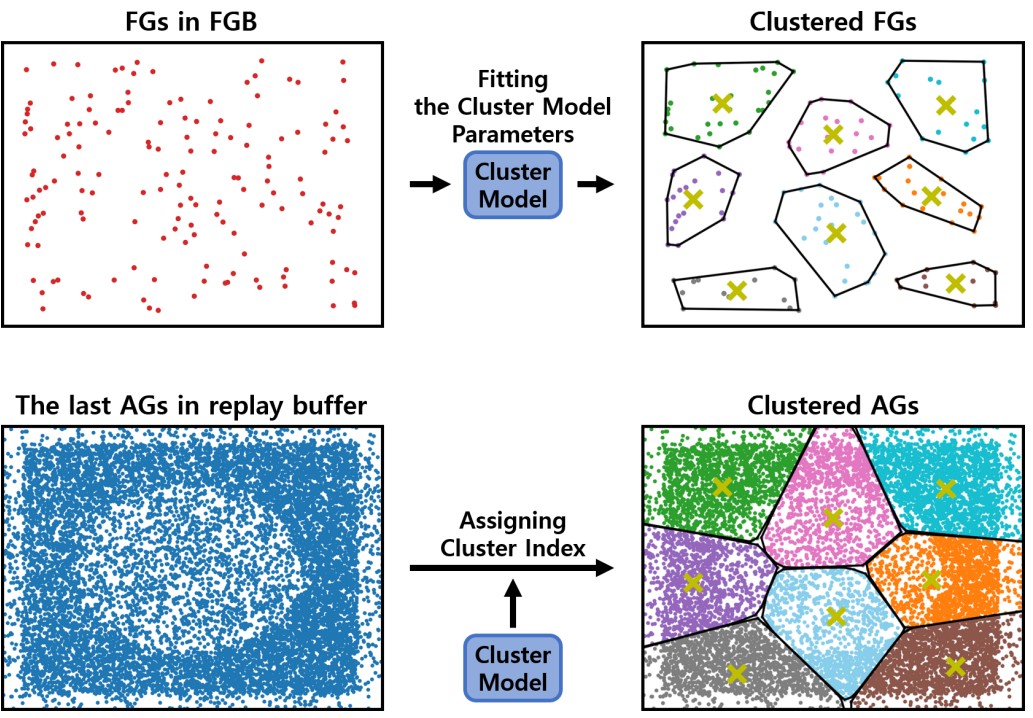

**Figure 3** **An example of fitting the cluster model and assigning cluster indices.** These two processes are sequential, with the fitted model being used for assigning cluster indices. The yellow 'x' denotes the centroids of clusters.

In short, the clustering process, which is the key idea of the proposed method, aims to increase the sampling probability of episodes whose last AGs are close to difficult-to-achieve FGs during training. A goal is considered an FG if the last AG in an episode differs from the original goal, indicating an unsuccessful episode. These FGs are stored in the FGB and are later used to fit the cluster model, which clusters the episodes in the replay buffer. The main concept of the proposed method can be expressed by introducing a cluster model $\mathbb{C}$ and a clustering-based sampling strategy $\mathbb{S}_c$ to HER. The pseudo-code for the clustering and sampling procedures of episodes and the proposed FAHER method are presented in Algorithm 1 and Algorithm 2, respectively.

# EXPERIMENTS

In this section, the experiment environment is described the experimental results of comparative evaluations, ablation studies, and three variations of the Slide task are provided.

## Experiment environment

Experiments are conducted on the continuous control tasks of the multi-goal environment used in _Plappert et al. (2018)_. The Fetch environment is developed by the OpenAI Gym (_Brockman et al., 2016_) using the MuJoCo physics engine (_Todorov, Erez & Tassa, 2012_). The performance of the proposed method is evaluated on three tasks within the Fetch

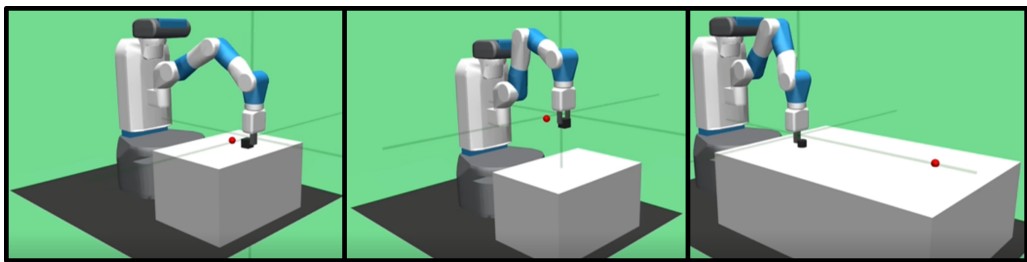

**Figure 4** **Illustrations of three tasks considered in experiments: Push, PickAndPlace, and Slide tasks.**

environment, involving a seven-degree-of-freedom robotic arm and an object placed on a table, as illustrated in Fig. 4. The three tasks are described as follows:

- Push task (FetchPush-v1): A goal location, a small red sphere in the figure, is randomly chosen in the 0.3 m × 0.3 m 2D space on the table surface within reach of the robot. The robot arm pushes the object (a box) to the goal location.
- Pick and Place task (FetchPickAndPlace-v1): A goal location is randomly chosen in the 0.3 m × 0.3 m × 0.45 m 3D space above the table. The robot arm picks the object (a box) with the gripper and places it at the goal location.
- Slide task (FetchSlide-v1): A goal location is randomly chosen in the 0.6 m × 0.6 m 2D space on the table surface in front of the robot, but out of reach of the robot. The robot arm slides the object (a puck) to the goal location.

In the three tasks, each episode consists of 50 timesteps. The episode is considered successful under the condition that the distance between the goal location and the object is less than a threshold value, five cm, in the last timestep.

For the experiments, DDPG is employed, where the actor and critic networks utilize a multi-layer perceptron architecture with rectified linear unit (*Nair & Hinton, 2010*) activation functions. The ADAM optimizer (*Kingma & Ba, 2014*) is used for the backpropagation algorithm to train both networks. The hyperparameters used in experiments are adopted from a paper presenting the experimental environment of HER (*Andrychowicz et al., 2017*).

## Experimental results

This subsection presents the experimental results of FAHER. The performance of the proposed method is evaluated in terms of the success rate during training and/or the final success rate after training. The figures illustrate the success rate during training, while the tables present the final success rate after training. The success rate during training is evaluated based on 20 test episodes, across a total of 200 training epochs. To ensure the robustness and reliability of the reported results, the entire sequence of training and evaluation is repeated using five different random seeds, and the results are averaged. The use of multiple random seeds helps in averaging out any potential fluctuations caused by random initializations and environment dynamics, thereby providing a more reliable measure of the performance.

In the figures, a solid line represents the average of the five success rates for each epoch, while the lower and upper boundary lines of the shaded area indicate the minimum and maximum success rates, respectively. To mitigate the granularity of the epoch-wise experimental results, the moving average of the past 20 success rates is calculated and plotted in the figures. The final success rate after training is calculated over 1000 test episodes using the model obtained after completing the 200 training epochs. The values reported in the tables for the final success rate are the averages calculated using five models trained with different random seeds, each evaluated over 1000 test episodes spanning 10 different random seeds.

To fit the cluster model to the FGs, K-means clustering algorithm is employed with the predefined number of clusters $k$ set to 8. The parameters governing the size of the FGB and the clustering cycle are both configured as 150. Whenever 150 new failed goals are accumulated in the FGB, the cluster model is updated. The implementation utilizes the scikit-learn library (*Pedregosa et al., 2011*), leveraging its built-in fit() function to fit/update the cluster model and the predict() function to assign clusters to AGs. The hindsight experience rate during sampling is set to 0.8 for all experiments.

Following the introduction of the clustering step for FAHER, a computational cost analysis is conducted to compare the training times between FAHER and traditional HER. Over 200 epochs, HER required approximately 94 min on a standard personal computer, whereas FAHER took around 112 min. However, this trade-off is balanced by the improvements in sample efficiency and success rates achieved through the clustering-based sampling strategy in FAHER. The following sections present detailed experimental results and analysis.

### Comparative evaluations

In the upper part of Fig. 5, the performance comparison between HER and FAHER is presented for three tasks. The lower part compares the performance of HER+EBP and FAHER+EBP. The "+EBP" notation indicates that the EBP algorithm, an existing algorithm designed to enhance sampling efficiency for HER, is used in conjunction. EBP is applied during the first uniform sampling step in Fig. 2, which involves sampling episodes from the replay buffer in the case of HER, and from the clustered episodes in the case of FAHER. In order to further understand the effectiveness of the proposed clustering-based sampling strategy, the increase in the ratio of PiE among sampled episodes is analyzed during training for both HER and FAHER. Throughout the training process, the sampled episodes are stored and evaluated. Since a key characteristic of PiE is that their AGs, if set as the original goals, are not achievable by the current RL policy, each AG of PiE is evaluated based on the current policy. The results showed that, on average, the ratio of PiEs increased by 9.82%, 94.61%, and 2.17% in the Push, PickAndPlace, and Slide environments, respectively. These increases contributed to the observed improvements in performance across tasks, highlighting the effectiveness of the clustering-based sampling strategy.

Figures 5A–5E demonstrate that the proposed method enhances the performance of both HER and HER+EBP. For the Push task, the proposed method reduces the number of epochs required to converge from 125 to 75 with HER and from 55 to 50 with HER+EBP.

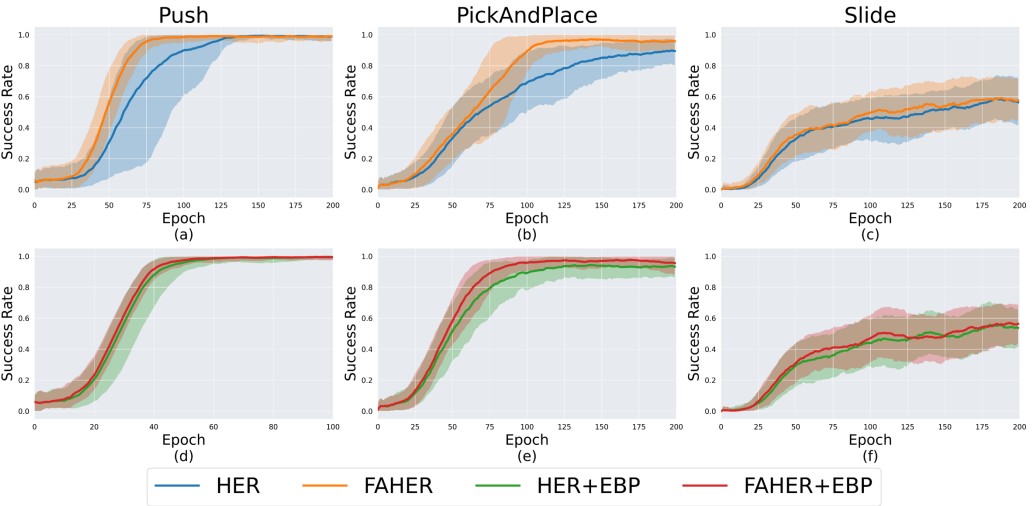

**Figure 5** Success rates obtained while training HER, FAHER, HER+EBP, and FAHER+EBP: (A–C) compare the performance of HER and FAHER; (D–F) compare the performance of HER+EBP and FAHER+EBP.

**Table 1 The final success rates of HER, FAHER, HER+EBP, and FAHER+EBP.**

| Method | Push | PickAndPlace | Slide |
|---|---|---|---|
| HER | 99.27 ± 0.19% | 91.97 ± 1.63% | 56.92 ± 2.15% |
| FAHER | 99.27 ± 0.37% | **97.00 ± 0.66%** | **59.10 ± 2.18%** |
| HER+EBP | 99.34 ± 0.33% | 93.64 ± 0.98% | 56.04 ± 1.95% |
| FAHER+EBP | **99.52 ± 0.17%** | 95.61 ± 0.87% | 58.38 ± 3.24% |

For the PickAndPlace task, the proposed method increases the maximum success rate during training by 4.48% and 3.29% compared to HER and HER+EBP, respectively. Specifically, Figs. 5D and 5E demonstrate that the proposed algorithm is suitable for use in conjunction with existing sampling algorithms. Table 1 shows the final success rates of HER, FAHER, HER+EBP, and FAHER+EBP. The proposed method improves the final success rate by up to 5.03% compared to that of the original method.

As shown in Figs. 5C and 5F, all four algorithms exhibit comparable performance, indicating the inherent difficulty of the Slide task itself. Due to the inherent difficulty of the Slide task, performance comparisons are not clearly visible. To address this, additional experiments are provided in 'Variations of Slide Task'.

### Ablation studies

The ablation study consists of experiments on the key methodological components of FAHER, which are the value of $k$ for the K-means algorithm, the size of the FGB, the inclusion or exclusion of FGs, and the clustering cycle.

In FAHER, the $k$ value for the K-means algorithm is set to 8. To evaluate the appropriateness of the $k$ value, experiments are conducted with $k$ values of 4, 8, and 16. In the proposed algorithm, $batch\_size/k$ episodes are sampled from each clustered

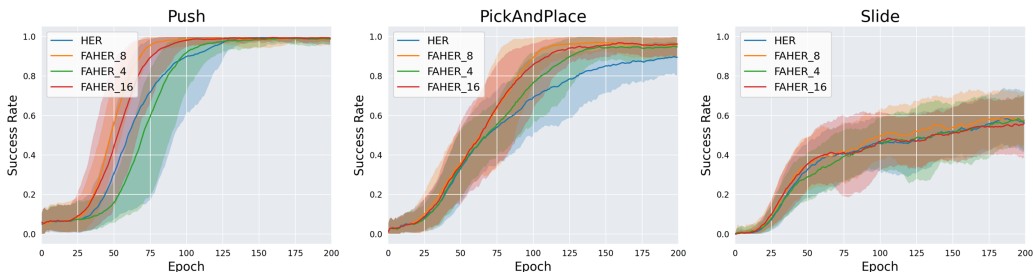

**Figure 6** **Success rates obtained while training HER, and FAHER with different *k*.**

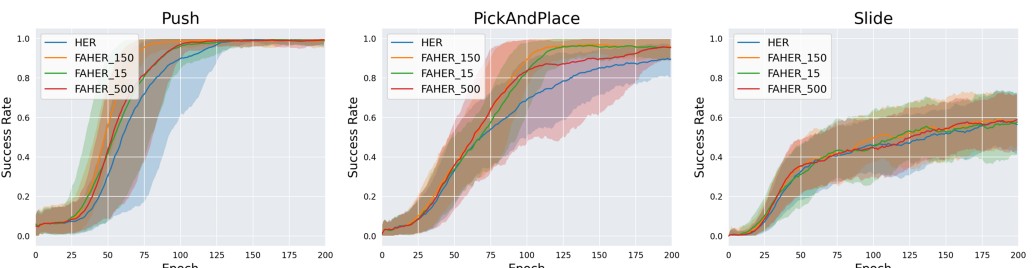

**Figure 7** **Success rates obtained while training HER, and FAHER with different sizes of FGB.**

buffer. Since *batch_size* is typically a power of 2, the test values for $k$ are chosen as powers of 2. Figure 6 compares the performance of HER and FAHER with different $k$ values. FAHER_ $i$ denotes $k = i$. For the Push and PickAndPlace tasks, FAHER_8 outperforms both FAHER_4 and FAHER_16. This result indicates that if $k$ is too small or too large, the sampling approach becomes similar to uniform sampling over the entire replay buffer.

The size of FGB is set to 150. To check the validity of 150 as the size of the FGB, experiments with different sizes of the FGB are conducted. FAHER with the FGB of size 150, 15, and 500 are named FAHER_150, FAHER_15, and FAHER_500, respectively. FAHER_150 is the same as FAHER used in other experiments. As shown in Fig. 7, FAHER_150 outperforms FAHER_15 and FAHER_500. This result suggests that the 150-size FGB is optimal for the cluster model of the proposed method. In contrast, the 15-size FGB is not sufficiently representative of the FGs for the RL model, and the 500-size FGB slows down the training of the RL model because it includes FGs from past RL policies.

The ablation studies on the number of clusters $k$ and the size of the FGB demonstrate that FAHER consistently outperforms HER, regardless of the hyperparameter values. This indicates that the proposed method enhances the performance of HER. As shown in the graphs, the success rates vary with different hyperparameter values, highlighting the importance of selecting appropriate values to achieve optimal performance.

The cluster model is fit to FGs obtained from the exploration of the RL policy. To assess the significance of the inclusion of FGs, experiments are conducted with HER using the clustering procedure without FGs (FAHER_woFG). In FAHER_woFG, the cluster model is fit to AGs in the replay buffer and assigns a cluster index to each episode. Across all three

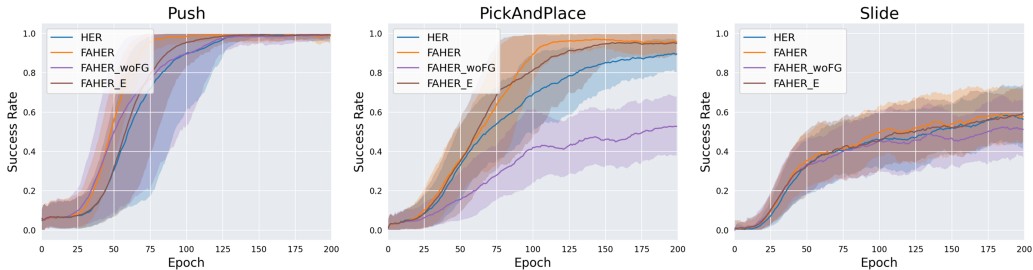

**Figure 8** Success rates obtained while training HER, FAHER, FAHER_woFG, and FAHER_e.

**Table 2** The final success rates of HER, FAHER, FAHER_woFG, and FAHER_e.

| Method | Push | PickAndPlace | Slide |
|---|---|---|---|
| HER | $99.27 \pm 0.19\%$ | $91.97 \pm 1.63\%$ | $56.92 \pm 2.15\%$ |
| FAHER | $99.27 \pm 0.37\%$ | $\mathbf{97.00 \pm 0.66\%}$ | $\mathbf{59.10 \pm 2.18\%}$ |
| FAHER_woFG | $99.15 \pm 0.26\%$ | $55.66 \pm 2.51\%$ | $49.47 \pm 4.92\%$ |
| FAHER_e | $99.15 \pm 0.25\%$ | $95.32 \pm 1.00\%$ | $57.85 \pm 1.31\%$ |

tasks, the results of FAHER_woFG are inferior to FAHER and even worse than HER, as indicated in Fig. 8 and Table 2. This outcome is attributed to the fact that when sampling the same number of episodes from each clustered buffer and one buffer has fewer experiences, the limited and worthless experiences are repeatedly sampled in FAHER_woFG.

The clustering cycle is set to 150, consistent with the size of the FGB, enabling the update of the cluster model with entirely new FGs. To evaluate the importance of this setting, an extreme case of using a short clustering cycle is compared with the proposed method using a cycle of 150. This extreme case involves setting the cycle to 1, meaning that the update of the cluster model and the procedure of clustering the episodes in the replay buffer are conducted in every episode (FAHER_e). Across all three tasks, as shown in Fig. 8 and Table 2, it is observed that the results of FAHER_e are either better or similar to HER but inferior to FAHER.

### Variations of slide task

In this section, three new variation tasks are defined, and the performance of HER and FAHER is compared for each task. Variation tasks are defined by modifying the goal space of the original Slide task, as depicted in Fig. 9. The distribution of red dots in the figure represents the goal space. The variation tasks are as follows: CloseSlide, which has a goal space with relatively close positions; FarSlide, which has a goal space with relatively far positions; and FarNarrowSlide, which has a narrow goal space located at a far position. In Fig. 10 and Table 3, the performance comparison between HER and FAHER is presented for three variation tasks. The proposed method improves the final success rate by 3.80%, 22.58%, and 7.09% compared to that of the original method for the CloseSlide, FarSlide, and FarNarrowSlide tasks, respectively. The variation tasks pose significant challenges due to the sparsity of successful experiences and narrow goal space. The low performance of

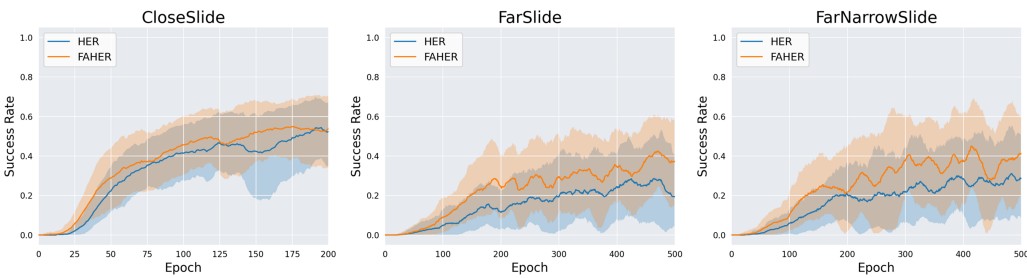

**Figure 9** **Illustration of the original Slide task and the three variations of the Slide task.** The black dot represents the initial positions of the object across 1,000 episodes, while the red dot indicates the goals for those episodes.

**Figure 10** **Success rates obtained while training HER and FAHER.**

**Table 3** **The final success rates of HER and FAHER for variation tasks.**

| Method | CloseSlide | FarSlide | FarNarrowSlide |
|--------|------------|----------|----------------|
| HER | 43.68 ± 9.17% | 14.26 ± 5.08% | 39.67 ± 6.92% |
| FAHER | **57.48 ± 2.02%** | **36.84 ± 11.95%** | **46.76 ± 4.50%** |

HER highlights the inherent difficulty of the task and serves as a rigorous test case for evaluating the effectiveness of the proposed method. The considerable improvements achieved by the proposed method in these tasks demonstrate its capability to enhance learning performance in the slide domain.

## CONCLUSION

This paper introduced Failed goal Aware Hindsight Experience Replay (FAHER), which improves HER by clustering episodes based on failed goals to increase the sampling of Potential-insights Episodes (PiEs). Experiments on robotic control tasks, including variation tasks, demonstrate that the proposed method can enhance sampling efficiency. In the Push task, FAHER reduces the number of epochs required to converge from 125 to 75 compared to HER. In the PickAndPlace task, it increases the final success rate by 5.03%. Notably, in the tasks related to the slide domain, the proposed method achieves an improvement of up to 22.58% in the final success rate. Additional ablation studies highlight the significance of the methodological components: the number of clusters, the size of the failed goal buffer(FGB), the inclusion of failed goals, and the clustering cycle.

Despite the promising results, the proposed method has several limitations. First, the performance of the proposed method is sensitive to hyperparameters, such as the number of clusters and the size of the FGB. These parameters were chosen based on preliminary ablation studies, which helped establish a solid baseline for comparison. Second, the K-means algorithm is used for the cluster model due to its simplicity and computational efficiency. Third, the lack of a direct comparison with recent HER-based methods is a limitation. While we attempted to implement these methods, achieving consistent performance proved challenging without stable publicly available code. Lastly, the experiments were conducted in relatively simple single-object environments to validate the fundamental principles of FAHER. Applying the method in multi-object or more complex scenarios could require additional complementary algorithms or sophisticated strategies, which were beyond the current scope due to implementation challenges.

To address these limitations, future research could explore adaptive mechanisms for dynamically adjusting the hyperparameters based on the learning progress of the RL policy. Additionally, investigating alternative clustering algorithms, such as density-based (*Ester et al., 1996*) or hierarchical clustering (*Murtagh & Contreras, 2012*), may help better capture the complexities of diverse environments. Simultaneously, future studies should aim to refine the implementation and conjunction of more recent HER-based methods to establish a broader comparison framework. This would allow for a more comprehensive understanding of the advancements in HER-based approaches. Further, evaluating FAHER in more complex multi-object environments would provide insights into its scalability and generalizability in real-world applications. To achieve this, combining FAHER with other complementary algorithms or strategic frameworks may be necessary to tackle the increased complexity of such tasks effectively.

### Funding
This work was supported by the Korea Institute of Energy Technology Evaluation and Planning (KETEP) grant funded by the Korea government (MOTIE) (No.20222020800190, Development and empirical study of a 600kW automatic charging system for customer convenience-based parking towers of 50 units capable of simultaneously charging 8 units for carbon-neutral acceleration). The funders had no role in study design, data collection and analysis, decision to publish, or preparation of the manuscript.

### Grant Disclosures
The following grant information was disclosed by the authors:
The Korea Institute of Energy Technology Evaluation and Planning (KETEP) grant funded by the Korea government (MOTIE): No. 20222020800190.
Development and empirical study of a 600kW automatic charging system.

### Competing Interests
The authors declare there are no competing interests.

## Author Contributions

- Taeyoung Kim conceived and designed the experiments, performed the experiments, analyzed the data, performed the computation work, prepared figures and/or tables, authored or reviewed drafts of the article, and approved the final draft.
- Taemin Kang performed the experiments, performed the computation work, prepared figures and/or tables, authored or reviewed drafts of the article, and approved the final draft.
- Haechan Jeong performed the experiments, analyzed the data, performed the computation work, prepared figures and/or tables, and approved the final draft.
- Dongsoo Har analyzed the data, authored or reviewed drafts of the article, and approved the final draft.

## Data Availability

The code is available at GitHub and Zenodo:

- https://github.com/ngng9957/Clustering-based-Failed-Goal-Aware-Hindsight-Experience-Replay

- Taeyoung Kim. (2024). ngng9957/Clustering-based-Failed-Goal-Aware-Hindsight-Experience-Replay: Clustering-based-Failed-Goal-Aware-Hindsight-Experience-Replay (FAHER). Zenodo. https://doi.org/10.5281/zenodo.14013517.

## Supplemental Information

Supplemental information for this article can be found online at http://dx.doi.org/10.7717/peerj-cs.2588#supplemental-information.

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
