# Peer review of "Clustering-based Failed goal Aware Hindsight Experience Replay"

_PeerJ Computer Science, doi:10.7717/peerj-cs.2588_

## Round 0.1 · original submission · Major Revisions

The authors must revise the manuscript carefully!

Reviewer 1 ·

Basic reporting

This paper addresses the low sample efficiency problem of the original HER algorithm. In the sampling step, it attempts to improve the sampling efficiency by leveraging some prior knowledge.
Experimental results show that in the basic Fetch series tasks, the proposed method can improve the sampling efficiency compared to the original HER algorithm (2017).
However, compared to the improved HER algorithm, the HER-EBP algorithm (2018), the improvement is not significant.
Moreover, this paper's method introduces two hyperparameters: the number of clusters k and the size of FGB. The performance is sensitive to these two parameters.

Finally, I do not believe that the introduction of FGB can effectively improve the sample efficiency of the HER series algorithms. The sample efficiency of the HER algorithm in manipulation tasks is mainly due to the consistent non-negative reward problem. For more details, please refer to a paper that is not mine: Bias-reduced hindsight experience replay with virtual goal prioritization.

However, as a student paper, its writing and workload are still suitable for publication in the current journal.

Score: 3/5

Experimental design

The experimental design is relatively complete, with comparative experiments and ablation studies that can comprehensively support the main points of the article.
However, the experimental scenarios are relatively simple, and the method has not been verified in more multi-object scenes, making it impossible to determine the universality of the hyperparameters.
Although the supplementary Slide experiment shows a significant improvement, it is essentially because the baseline performance is too poor, making the improvement appear substantial.

Score: 3.5/5

Validity of the findings

I am not sure about the author's description of using five random seeds in Experimental Results: the sequence of training followed by evaluation is repeated with 5 different random seeds.
* * *
The evaluation and design of the experimental results are similar.

Additional comments

1. It would be better to describe the first two figures in more detail, as I still have some confusion about this part of the content.

2. Why is the efficiency low for successfully sampled samples?

3. Hasn't this past experience sample just undergone goal relabeling and obtained a valid positive reward? Why would it also be low?

4. In the paper's goal relabeling strategy, is the Last strategy or the Future strategy used? If it's Last, the sample efficiency itself is low and not suitable for the HER algorithm.

5. The code is written quite standardly, although I haven't run it, most of the hyperparameters are inherited from the original HER algorithm.

6. The goal of the failed rounds in the code is more clear to read: compare the last ag of the round with the original g, if it exceeds the threshold of 5cm, it is considered that the goal has failed to be completed. This goal g is then stored in fgb. Then these failed g's are clustered. The authors believe that the sampling probability of these difficult-to-achieve failure goals should be increased.

7. However, I actually think this is only a local feature of single-object manipulation tasks. For complex multi-object manipulation tasks, sampling these difficult-to-achieve failure goals may not help improve the policy.

Cite this review as

Reviewer 2 ·

Basic reporting

Language and Clarity:
The English language used in the manuscript requires significant improvement to ensure clear communication to readers. For examples, lines 34-36, 55, 59, 61, 99, among others, need revision. It is recommended that the authors seek assistance from a native English speaker or a professional English editing service to enhance the overall readability.

Organization and Focus:
The manuscript could benefit from a more focused presentation, particularly by organizing the background on reinforcement learning and Hindsight Experience Replay (HER) in a separate preliminary section. This will enhance the readability and structure of the paper.

Supporting References:
I suggest adding appropriate references to support the claims made, especially in line 45 regarding the disadvantages of reward design.

Related Work:
The manuscript lacks a comprehensive comparison with recent work in HER. There has been a lot of follow-up research on HER, so it is necessary to detail how it differs from the proposed method and compare it to more recent work including:
- Episodic self-imitation learning with hindsight
- Self-Imitation Guided High-Efficient Goal Conditioned Reinforcement Learning
- Hindsight expectation maximization for goal conditioned reinforcement learning
- Visual Hindsight Self-Imitation Learning for Interactive Navigation

Figure Clarifications:
The meaning of the question mark in Figure 1 is unclear and requires an explanation. Additionally, the paper’s contribution (number 3 in the figure) should be explained in greater detail.

Methodological Clarifications:
The distinction between hindsight experience (HE) and past experience is not sufficiently clear,
particularly in differentiating between AG (Achieved Goal) and FG (Failed Goal). Both are derived from failed experiences, making it difficult to understand their differences in this context.

Experimental design

Sampling and Clustering Analysis:
If the intent behind the method is to increase sampling diversity through clustering, the analysis should demonstrate whether this goal was achieved. I emphasize the need for an analysis of the differences in sampling variations. Furthermore, the manuscript should include comparisons with more recent methods, as only HER is used for comparison, which is considered outdated. The experimental environment also needs to be more varied.

Clustering Demonstration (Figure 3):
The clustering shown in Figure 3 lacks well-defined boundaries, raising concerns about whether well-formed clusters were achieved. This needs further clarification and validation.

Applicability of the Proposed Method:
Questions are raised regarding the applicability of the proposed method for the second and third uniform samplings with the episode batch. If applicable, its performance should be evaluated and discussed.

Performance Analysis: The performance differences observed are minimal and often fall within the margin of error, making it difficult to claim any significant advantage. Additionally, I question why FAHER_e, which performs clustering at each step, does not outperform FAHER_150 as expected.

Validity of the findings

Strengths: The authors present a sampling method that potentially improves HER performance, demonstrating that sample-efficient learning is possible.

Conclusion and Future Directions: The conclusion lacks any mention of future research directions or the limitations of the proposed approach.

Additional comments

Code Release: The authors are commended for releasing the code on GitHub. However, the documentation is insufficient, making it difficult for users to utilize the code effectively.

Cite this review as

Reviewer 3 ·

Basic reporting

- The paper is well-written with clear, unambiguous language. However, some sections, particularly the methodology, could benefit from clearer explanations regarding technical terms and processes, especially for readers who are not experts in the field of reinforcement learning.

- The structure follows standard academic conventions. The introduction, methods, experiments, and conclusions are well-organized, with sufficient background provided in the introduction to establish the context of the work.

- The figures are clear, well-labeled, and relevant to the content. The visual representations of the tasks and the algorithm are helpful in understanding the method. Raw data is adequately presented, and the tables are effective in conveying the performance comparisons.

- The references are appropriate and relevant to the field. The paper builds upon prior work, especially Hindsight Experience Replay (HER), and situates the proposed FAHER method within the broader literature.

Experimental design

- The research question is well-defined and addresses a clear gap in improving sampling efficiency in Hindsight Experience Replay (HER) by focusing on failed goals (FGs) using a clustering-based method. This is a relevant and meaningful area of inquiry in reinforcement learning.

- The proposed methodology is sound but could benefit from additional explanation. For instance, the clustering model used for sampling is critical to the FAHER method, but the reasoning behind some parameter choices (e.g., the number of clusters, FGB size) could be more thoroughly justified. The use of K-means clustering is appropriate but could benefit from more detailed analysis regarding its limitations or alternative approaches.

- The methods are described with enough detail that they could be replicated, but providing additional pseudocode or more detailed algorithm descriptions (especially for the clustering process) would improve clarity for replication purposes.

Validity of the findings

- The results are robust, supported by comparisons with baseline HER methods, as well as additional variants like HER+EBP. The experiments are run across multiple random seeds, ensuring the findings are statistically sound.

- The findings show that FAHER consistently improves upon HER, particularly in tasks with more complex goal structures (e.g., the Slide task). The paper also includes ablation studies that explore the impact of key methodological components, such as the clustering cycle and FGB size, which add validity to the proposed method.

- The paper lacks a detailed discussion of the limitations of FAHER. It would be beneficial to discuss potential downsides, such as computational overhead from clustering or scenarios where the method might not perform as well as HER.

Additional comments

Strengths:

+ The paper introduces a novel approach (FAHER) that meaningfully extends HER, particularly in sparse reward settings.

+ The experimental results are strong, showing clear improvements over baseline HER, especially in challenging robotic tasks.

+ The ablation studies are thorough and provide valuable insights into the key parameters and design choices of the method.

+ The figures and tables are well-designed and help in visualizing the results.

+ The language is generally clear and professional.

Weaknesses:

Major:

- The explanation of the clustering process and the rationale for the chosen parameters (e.g., number of clusters) could be more detailed.

- While the results are promising, there is a lack of exploration into the computational cost or trade-offs of using FAHER versus traditional HER methods.

- The paper would benefit from a discussion of potential limitations or scenarios where FAHER might not offer improvements.

Minor:

- Some sections, particularly the introduction and conclusion, could be more concise.

- The paper does not mention future research directions, which could provide readers with a sense of how the work can be further developed.

Cite this review as

---

## Round 0.2 · accepted · Accept

The paper was well improved and I believe it is now Acceptable.

Reviewer 1 ·

Basic reporting

I was surprised to see that our three reviewers raised 45 comments in total, and overall, the authors have provided detailed responses and supplements.

I have reviewed all the review comments and the new version of the manuscript with 4 hours.

During my previous reading, due to issues with the presentation, I indeed did not understand the method's motivation. This version has made it somewhat clearer to me.

The current version basically meets the publication requirements of this journal. However, I would like to offer some additional suggestions to the authors.

In Figure 1, the second scenario actually represents a path to obtain effective non-negative rewards, which can be referenced in the paper "Relay Hindsight Experience Replay: Self-guided continual reinforcement learning for sequential object manipulation tasks with sparse rewards". The key insight here is that in GCRL, the achieved goal needs to be changed by the agent's actions, and exploration guidance is necessary to achieve relatively high sample efficiency.

Comparatively, the aforementioned work still doesn't fully utilize the data, which corresponds to the third scenario in Figure 1. The goal space data augmentation can be further enhanced, as referenced in "Invariant transform experience replay: Data augmentation for deep reinforcement learning". The key insight here is that besides sampling hindsight goals from agent-explored achieved goals, sampling can be done across the entire goal space. Crucially, the sampling size here is substantial, covering the entire object space, such as within 5cm of the achieved goal + the entire goal space.

Based on these literature and my experimental experience, for Fetch series tasks, using the original HER's Future strategy can achieve 4x data augmentation. Additionally, it's possible to achieve 16x more data augmentation, meaning there can be 1 original exploration sample + 20 augmented samples. I haven't published these experimental data, and after completing my Ph.D., I'm no longer interested in pursuing this. However, the authors are welcome to try this approach.

Most importantly, such data augmentation can reduce sensitivity to hyperparameters.

Finally, I suggest moving away from HER and Fetch series tasks as they are outdated. After publishing this paper, I recommend pursuing more interesting research directions.

Experimental design

no comment

Validity of the findings

no comment

Additional comments

no comment

Cite this review as

Reviewer 2 ·

Basic reporting

The revised manuscript has primarily addressed my comments from the initial review.
However, Figure 1 still lacks the "green arrow" referenced in the caption and fails to convey the study's motivation effectively. Furthermore, incorporating numerous comments into the manuscript may have disrupted the overall flow in certain sections. I recommend the authors carefully review and refine the entire manuscript before submitting the final version for publication. Addressing these issues would significantly enhance the quality of the paper.

Experimental design

no comment

Validity of the findings

no comment

Additional comments

no comment

Cite this review as

Reviewer 3 ·

Basic reporting

The authors have addressed my raised concerns.

Experimental design

The authors have addressed my raised concerns.

Validity of the findings

The authors have addressed my raised concerns.

Additional comments

NA

Cite this review as